# Influence of the Reinforcement Structure on the Thermal Conductivity and Surface Resistivity of Vinyl Ester Composites Used on Explosion-Proof Enclosures of Electrical Equipment

**DOI:** 10.3390/ma15155190

**Published:** 2022-07-26

**Authors:** Małgorzata Szymiczek, Dawid Buła, Jacek Koczwara

**Affiliations:** 1Department of Theoretical and Applied Mechanics, Silesian University of Technology, 44-100 Gliwice, Poland; 2Department of Electrical Engineering and Computer Science, Silesian University of Technology, 44-100 Gliwice, Poland; dawid.bula@polsl.pl; 3Ośrodek Pomiarów i Automatyki S.A., 41-800 Zabrze, Poland; 4Mastermodel FHU Jacek Koczwara, 43-300 Bielsko-Biała, Poland; jacek.koczwara@mastermodel.pl

**Keywords:** vinyl ester laminate, thermal conductivity coefficient, surface resistivity, flexural strength, composite enclosures

## Abstract

The study aimed to evaluate the influence of structure (type and material) on thermal properties (thermal conductivity, diffusivity) and surface resistance of composites used for explosion-proof enclosures of electrical devices. The matrix was a graphite-modified flame retard vinyl ester resin. As part of the work, 4 structures of composites reinforced with glass fabric, glass mat, and carbon fabric were tested. The composites were prepared by hand lamination with a vacuum. A methodology for indirectly determining the thermal conductivity coefficient was developed, taking into account the geometry of the explosion-proof enclosures. Thermal diffusivity, surface resistivity, flexural, and inter-layer shear strength were tested. The specific strength of the composites was determined. The highest properties were shown by the composite with carbon reinforcement, but for economic reasons, the enclosure was made with glass fabric. In the final stage, the model of the composite explosion-proof enclosure was designed and manufactured, followed by quality verification using pressure tests. The presented results are the next stage of work, the aim of which is to design and manufacture explosion-proof enclosures for electrical devices made of polymer composites. Based on the obtained results and economic factors, a composite with an S1 structure was selected for the preparation of the enclosure. It was found that the combination of graphite-modified vinyl ester resin and triaxal 550 g/m^2^ glass fabric allows for high internal pressure resistance. (8 bar). The proposed solution will allow for reducing the weight of explosion-proof enclosures while meeting the assumed operational requirements.

## 1. Introduction

Construction composites, due to their properties, especially their specific strength, are used in various, often very demanding applications. The areas of application require the synergy of strength, thermal and electrical properties, resistance to fire, and even their cross-section variability. A particularly interesting area to apply is mining, means of transport, and defense industries. The weight of the structure translates into mobility, ease of transport, and assembly, reducing the emission of harmful substances to the environment.

Electrical devices generate a lot of heat, therefore the basic property of their enclosures is to dissipate heat. Therefore, metal alloy or cast-iron enclosures are most often used in the mining industry, which allows for quick heat dissipation, but they are heavy. An alternative to the currently operating electrical equipment enclosures is made of composite polymer materials with high strength properties and low density (high specific strength). However, composites must have a high coefficient of thermal conductivity, low surface resistivity below 10^9^ Ω, and fire resistance (V-0 flammability class according to UL 94 (Standard for Safety of Flammability of Plastic Materials for Parts in Devices and Appliances) [1]. Explosion-proof enclosures are also important in operating conditions: temperature from −20 °C to +60 °C, pressure from 80 kPa (0.8 bar) to 110 kPa (1.1 bar); and air with normal oxygen content, usually 21% vol. [2].

The resins are characterized by low fracture toughness, which can be improved by introducing fillers. Modification of resins may concern various operational properties, depending on the fillers used [3,4,5,6,7]. The operational loads of the composite determine the choice of reinforcement [6,8,9,10]. Additionally, for composites operating under high thermal loads, heat sinks with a large heat exchange surface are used [11]. Due to the cold spark phenomenon, aluminum heat sinks cannot be used in the mining industry. Radiators are selected depending on the medium and working environment [12].

Functional fillers improve stiffness, fracture toughness, thermal conductivity, heat deflection temperature, reduce shrinkage, void spaces, and good appearance of composites [10], but change the curing conditions. In the literature, there are many different solutions depending on the application environment, increasing the thermal conductivity coefficient. Graphite, carbon fiber, carbon nanotubes [13], silicon carbide [14], aluminum nitride [15], aluminum oxide [4], copper oxide [16], boron nitride [17], metallic fillers [18] are used here. However, most of the work concerns the modification of epoxy resin. Burger et al. [13] showed, that defects in the crystalline structure inevitably result in phonon scattering, i.e., a decrease in thermal conductivity. Change in the linearity or regularity of the morphological aspect of the filler leads to a decrease in intrinsic thermal conductivity. The influence of fillers on properties is modeled using various theories [6,8,19]. The modeling process is more complicated for hybrid composites, in which the resin is modified by fillers (graphite, carbon non-tubing, copper), but the main reinforcement is fabrics (glass, carbon, aramid). In the work [7], graphite was introduced into the vinyl ester resin to increase the thermal conductivity and decreased the surface resistivity, but it changed the curing conditions. The change in the hardening time under the influence of graphite was shown in the studies by Du and Fang [20] and Jaswal and Gaur [21]. The properties of composites are determined by the characteristics of the components, composition, structure, and the fiber/filler-matrix interaction [6,8]. The type of reinforcement affects the properties of the composite. El-Hage et al. [22] found that the thermal conductivities in the plane of reinforcements are slightly greater than in the thickness. Zhou et al. [9] developed methods of structure optimization to increase the thermal conductivity of plain-weave textile composites. The distribution of heat in a manufactured part leads to inner stress and microcracks [23].

It is important to properly design a relatively cheap and lightweight composite for explosion-proof electrical enclosures which meets the requirements of the extractive industry. The research presented in the paper is the next stage of the project aimed at the demonstrator of the composite explosion-proof enclosure for electrical devices that generate a significant amount of heat.

The work aimed to assess the impact of the composite structure on surface resistivity, thermal conductivity, and selected strength properties at the V-0 flammability class according to UL94 [1]. The matrix was a modified vinyl ester resin. The tested properties were determined by the potential application of the developed structures to light enclosures of electrical devices operating in explosion hazard conditions. In the final stage, the enclosure was designed and subjected to verification tests. The research program included:development of the composite structure and manufacturing technology (taking into account the potential application),conducting thermal tests,determination of surface resistivity,determination of strength characteristics (flexural strength, flexural modulus, strain, and interlayer shear strength),verification pressure tests of the enclosure.

## 2. Materials for Research and Methodology

### 2.1. Materials for Research

The tests were carried out on composites reinforced with carbon quadriaxial fabric (Selcom SRL, Fregona, Italy), glass triaxial (Changzhou Pro-Tech Trade Co., Changzhou, China), and glass mat (Jushi Group Co., Ltd., Tongxiang City, China). The reinforcement material was selected in such a way as to minimize the property anisotropy. The arrangement of layers, designed with heuristic techniques (Delphi method) is summarized in Table 1. The thickness of the laminates was 7 mm ± 0.2 mm.

The matrix was a modified vinyl ester resin (POLIMAL-VE-11 MAT, Sarzyna, Nowa Sarzyna, Poland) based on brominated bisphenol-A. cured with a system of 1% cobalt accelerator (1% concentration) and 2% low-reactive methyl ethyl ketone peroxide (LUPAROX K -12 G). The resin was modified with flake graphite MG 394 (grain size (flake) < 45 µm) and MG 1596 (grain size (flake) < 10 µm) (Sinograf, Toruń, Poland); thermal conductivity is from 140 to 233 W/mK, depending on the character. The applied modification aimed to improve thermal conductivity and reduce surface resistivity. Table 2 shows the properties of the matrix material. The resin modification technology is described in [7].

Due to technological conditions, a concentrate was first prepared in the form of a paste made of vinyl ester resin and a mixture of modified graphite. The weight mix ratio between flake graphite MG 394 and MG 1596 was 3:1. Graphite was produced by Sinograf, Toruń, Poland. The resin concentrate was prepared on a High-Speed Dissolver Dispernat LC 30 mixer (VMA-Getzmann GMBH, Reichshof, Germany) equipped with discs (diameter of 50 mm). Mixing speed of 1000 rpm for 1 h. Mixing ratio 1:1 weight. In industrial conditions, the prepared concentrate was introduced into the resin (mix ratio: 6:4 parts by weight) using a 90 mm COLLOMIX LX 90 S mixer (Collomix GmbH, Gaimersheim, Germany) for 10 min at a speed of 2000 rpm. Curing system of 1% cobalt accelerator (1% concentration) and 2% low-reactive methyl ethyl ketone peroxide (LUPAROX K -12 G) was added. This ensured an appropriate degree of homogenization of the matrix material, which allows for the elimination of defects in the form of agglomerates, improper wetting of the fibers, and voids leading to delamination. The designed composites with dimensions of 500 mm × 500 mm were made by hand lamination with a vacuum. Figure 1 shows the laminating of composites on a table with an even and stable heating system. Composite panels were made by Mastermodel Sp. z o.o.

After cross-linking, the prepared composites were conditioned on a heated table at the temperature of 60 ± 2 °C for 2 h. After the heaters were turned off, the composite cooled along with the table. The conditioning time was 48 h. Test samples were cut by a water jet (PREMIUM 2D, WS3015/133302 FLOW model 7/3633-1 produced by STM STEIN-MOSER GmbH (STM Waterjet GmbH, Wasserstrahl-Schneidanlagen, Eben im Pongau, Austria). The cross-sectional view of the prepared samples is presented in Figure 2. Microscopic images were taken by the Zeiss SteREO Discovery stereo microscope (Carl Zeiss AG, Oberkochen, Germany). There are significant differences in the structure between the composites reinforced of glass fabric and mat (S1—Figure 2a, S2—Figure 2b, S3—Figure 2c) and carbon (S4—Figure 2d). This has a direct impact on the tested characteristics.

Before the implementation of the assumed test program, the samples were conditioned in forced-air dryers at a temperature of 80 ± 2 °C, for 4 h.

### 2.2. Thermal Properties Research

The thermal conductivity was determined according to the formula:(1)$$\lambda =d_{c}c_{p}\rho$$
where: *d*_c_—thermal diffusivity [mm^2^/s] *c*_p_—specific heat [J/gK], ρ—density [g/cm^3^].

Determining thermal conductivity was performed experimentally: thermal diffusivity, density, and the specific heat value were calculated based on the specific heat of the modified resin determined experimentally [7] and the specific heat of glass and carbon taken from the literature [24]. The thermal conductivity was selected analytically for mean values.

Thermal diffusivity was determined by the active thermography method, in which thermal activation was carried out with the use of an infrared radiator. The thermal response of the object as a function of time to the stimulation of the system with an external heat pulse was determined on the surface opposite to the heated one. The plate was mounted in an insulated holder allowing to eliminate heat conduction at a distance of 1 m. This ensured homogeneous conditions for thermal activation. The temperature distribution on the surface was registered by the FLIR A615 camera with the IRcontrol software (FLIR Systems, Inc, Wilsonville, OR, USA). Samples with dimensions of 500 mm × 500 mm × 7 mm were heated for 40 ± 2 s. The analysis values are the arithmetic mean of 5 measurements. The research did not take into account heat exchange with the environment and climatic conditions. Diffusivity was determined by the relationship [25]:(2)$$d_{c}=1.38\frac{g^{2}}{\pi^{2}t_{1/2}}$$
where: g—the thickness (mm) and *t*_1/2_—half the time to maximum temperature in seconds.

The density of the prepared composites was tested on an analytical balance (Ohaus Adventurer Pro, OHAUS Europe GmbH-Nänikon, Greifensee, Switzerland) equipped with a hydrostatic density measurement set, following ISO 1183-1 [26]. The samples were cut from the prepared composite panels with dimensions of 20 mm × 20 mm. The test consisted of measuring the mass of the sample in the air, and then in demineralized water of known density (ρ_w_ = 0.998 g/cm^3^). Based on the obtained results, the density ρ (g/cm^3^) of the samples was determined by the following formula:(3)$$\rho =\rho_{w}\frac{m_{1}}{m_{1}-m_{2}}$$
where *m*_1_ and *m*_2_—the sample mass in air and demineralized water, respectively (g). The density value is the average of five measurements. The tests were carried out at a temperature of 20 ± 2 °C and a humidity of 50 ± 5%.

The specific heat of the *c*_pc_ composite was determined analytically, with mixture rule:(4)$$c_{\mathrm{pc}}=c_{\mathrm{pm}}V_{m}+c_{\mathrm{pf}}V_{f}$$
where: *c*_pm_—specific heat of matrix [J/gK], *c*_pf_—specific heat of fibres [J/gK], *V*_m_—resin content [g], *V*_f_—fibres content [g].

The rule of mixtures assumes, i.a., that the material is isotropic, the adhesion between the filler and the matrix is not critical to the properties of the composite, no air bubbles and other defects. It is used primarily for materials reinforced with particles, and short fibers. Rule of Mixtures can be used in practice to determine the approximate properties of the composite along the continuous fibers despite the geometric inaccuracies of the model, but for the properties of the transverse direction it generates a problem resulting from the cooperation of the fiber with the matrix, which affects the state of deformation and stress [27]. According to [8], the thermal conductivity determined to follow the rule (4) gives a lower limit of the transverse thermal conductivity. Sherin et al. [28] used the mix-nin rule to define the dielectric and thermo-mechanical properties of Sm_2_Si_2_O_7_ filled polyethylene and polystyrene composites for microwave packaging. Omidi et al. [29] applied the mixture rule to the prediction of the mechanical characteristics of carbon nanonanotube/epoxyposites taking into account the shape of the fillers. The rule of mixture allows for predicting specific heat dependence of HDPE–Ag nanocomposites as a function of filler mass. However, the basic models of thermal conductivity underestimate the values for low and high filler content [30].

The weight content of reinforcement was determined by the method of burning composite samples with a muffle furnace at a temperature of 300 °C to a constant weight according to ASTM D3171-22 [31]. The weight difference was determined by the analytical balance (Ohaus Adventurer Pro, OHAUS Europe GmbH-Nänikon, Greifensee, Switzerland). Three samples with dimensions of 20 mm × 20 mm, cut from the central part of the plate, were subjected to testing to eliminate the influence of boundary conditions.

The specific heat of the matrix was determined in [7] by differential scanning calorimetry method by ISO 11357-4: 2013 [32] using the DSC 1 Star System apparatus by Mettler Toledo (Columbus, Ohio, OH, USA). The average value of three measurements was adopted for the calculations. The specific heat of the modified vinyl ester resin of 2.46 J/gK was assumed to determine the thermal conductivity. The specific heat of carbon fibers is 0.800 J/gK and glass fibers 0.45 J/gK [24].

In order to determine the impact of the simplifications used in the rule of mixtures for determining specific heat, it was also determined using the colorimetric method. The calorimetric method is based on the first law of thermodynamics and heat balance. The specific heat was determined for three samples with dimensions of 20 mm × 40 mm × 7 mm cut from laminate boards described in Section 2.1. It was calculated according to the formula:(5)$$c_{\mathrm{pc}2}=\frac{\left(m_{\mathrm{cal}}c_{\mathrm{cal}}+m_{\mathrm{water}}c_{\mathrm{water}}\right)\left(t_{e}-t_{i}\right)}{m_{s}\left(t_{b}-t_{e}\right)}$$
where: *m*_cal_—calorimetric mass [g], *c*_cal_—calorimetric specific heat = 0.896 [J/gK], *m*_water_—water mass [g], *c*_water_—water specific heat = 4.19 [J/gK], *t*_e_—end temperature [K], *t*_i_—initial temperature [K], *t*_b_—boiling point [K].

### 2.3. Surface Resistivity

The surface resistivity tests were carried out on composite panels with dimensions of 500 mm × 500 mm × 7.2 mm before cutting the samples (Figure 3a) on the upper surface, after removing the delaminating fabric, which allowed to eliminate the removal of the outer resin coating. Before the tests, the composite was degreased with ethanol. Figure 3b shows the test stand. The test was carried out using the static method by IEC 61340-2-3 [33] on the RS TR04 device (Robs SMT sp. z o.o., Bydgoszcz, Poland). The sensor measures the surface resistivity in the range: of 10^3^ ÷ 10^12^ Ω. It is equipped with two measuring probes weighing 2.33 kg and a diameter of 63.5 mm, allowing RTT (Resistance Point-to-Point) and RTG (Resistance Surface-to-Ground) measurements. The measurement time was 15 s. The tests were carried out at the temperature of 20 ± 2 °C and humidity of approx. 40 ± 5%. For each sample, 5 measurements were conducted.

### 2.4. Mechanical Tests

Flexural strength tests were carried out by ISO 178 [34] and ISO 14125 [35], on five samples with dimensions of 130 mm × 15 mm × 7 mm. The tests were performed on a Shimadzu AGX-V 10 kN D testing machine cooperating with the TrapeziumX-V software (ver. 1.3.4, Shimadzu Corporation, Kyoto, Japan) and equipment for three-point flexural. The machine and the software were manufactured by Shimadzu Corporation, Kyoto, Japan. The support span was assumed 16 × the sample thickness, so is 112 mm. The radius of the support pins was 5 mm. The test speed was 2 mm/min. Ambient temperature 20 ± 2 °C, air humidity 60%. It has been specified:flexural strength:
(6)$$\sigma_{f}=\frac{3FL}{2bh^{2}},$$
where: *F*—load [N], *L*—support spacing [mm], *b*—sample width [mm], *h*—sample thickness [mm].
flexural modulus:
(7)$$E_{g}=\frac{L^{3}}{4bh^{3}}\left(\frac{\Delta F}{\Delta s}\right),$$where: Δs—deflection increment [mm], ΔF—load increment [N].

Tests of inter-layer shear strength σ_s_ were carried out by ISO 14130 (short beam method) [36]. For each measuring point, 5 samples measuring 70 mm × 15 mm × 7 mm were tested. The tests were performed in the same conditions in which the flexural strength was determined. The support span was 35 mm. The test speed was 1 mm/min. The interlayer shear strength was calculated from the relationship:(8)$$\tau =\frac{3F}{4bh}$$

## 3. Results and Discussion

For the S1–S4 composite structures, tests of surface resistance, mechanical tests, and thermal properties tests were carried out. The mechanical strength was also confirmed by pressure testing of the physical model of the enclosure.

### 3.1. Thermal Conductivity

In the first stage, the content of reinforcing fibers was determined using the firing method. Figure 4a shows the fired composites. Figure 4b shows the averaged results of the modifier’s content–graphite and reinforcement. The research was conducted by the methodology described in Section 2.2.

It was observed that the lowest reinforcement content was for the mat reinforced composites (S3), which is related to the structure of the reinforcement itself. The reinforcement and graphite content for the structures S1 and S4 depends on the weight of the reinforcement. The thicker the fabric, the more difficult it is to attain good impregnation, as there are more voids and bubbles. For the difference in fabric grammage of 250 g/m^2^, the content of dry fillers (fabric and graphite) was 7%. The density is closely related to the content–Figure 5.

Based on the data and the determined value of modifiers (reinforcement and graphite), the approximate specific heat value (Formula (4)) was calculated and compared to the determined calorimetric method (Formula (5)). The results are shown in Figure 6.

The highest specific heat values are shown in the case of carbon fiber-reinforced composites. This is due to the specific heat of carbon, which is about 1.8 times higher than glass. The specific heat for the S3 structure was 1.35 J/gK. It results from the content of modified resin, whose specific heat is 2.46 J/gK, and the structure (weave) of the reinforcement. The specific heat determined by the calorimetric method is approx. 10% higher for composites with the S4 and S3 structures, and for the S2 and S1 structures by approx. 5%. Structure S1 had the lowest specific heat, which is related to the content of reinforcing fibers, their material property, and type. It can be concluded that the described procedure of determining the specific heat by the mixture method gives a good approximation of the actual specific heat.

Resin impregnates the glass mat well, which reduces the mechanical properties, but has a positive effect on thermal properties. The temperature distribution maps for the S4 and S1 structures are shown in Figure 7. The fabric-reinforced composites (S1 and S4) were compared. Differences in the thermograms can be observed, suggesting the presence of structural defects. The temperature changes in time are presented in Figure 8. The recorded changes are the average temperature from the area of approx. 100 mm^2^.

The analysis of thermograms showed that the process of heating and heat dissipation for carbon-reinforced composites is faster. Already after 45 s, significant changes in thermograms were observed. The S2 structure showed no temperature drop even after 50 s.

The thermal diffusivity was determined based on temperature changes in time during heating (Figure 9). The sample with the S4 structure (carbon fabrics) was heating the fastest (10.52 s), and in the final stage, a minimum temperature drop was observed. S4 also showed the highest temperature: 25.57 °C. The difference between the initial and the highest temperature was 3.48 °C. S3 structure reached the stabilized temperature in 17.71 s. The difference between the initial and maximum temperature was 3.19 °C. It was determined by the properties of the modified matrix. The temperature for the S1 and S2 structures stabilized the longest: 23.2 s. It is related to the type and properties of the reinforcement. Structures S1 and S2 have the lowest specific heat and the highest glass content. The higher the glass reinforcement content, the lower the specific heat. In addition, the glass fabric used is relatively difficult to impregnate, which has an impact on thermal diffusivity.

The determined thermal characteristics were used to determine the thermal conductivity coefficient. Results are shown in Figure 10.

The determined thermal conductivity coefficient is the effect of a specific thermal diffusivity, density, and specific heat. The highest coefficient of thermal conductivity was recorded for S4. The thermal characteristics of the reinforcing material have the greatest impact on the tested thermal properties. Carbon reinforcement is characterized by the highest thermal conductivity coefficient with the lowest composite density. The observed differences between the thermal conductivity determined from the specific heat, usually calculated in the rule of mixtures, and that determined by the calorimetric method, result from the structure of the composite but also can be related possible defects in the structure or the voids content.

### 3.2. Surface Resistivity

Table 3 shows the results of the surface resistivity. The presented results are average values of 5 measurements, and the differences between the individual measurements are within the error limits.

The structures S1, S2, and S3 are characterized by a surface resistivity at the level of 3 × 10^3^ Ω. For the S4 structure, it is below 10^3^ Ω. It can be concluded that the type of reinforcement is less important than the type of material (glass or carbon). Modification of vinyl ester resin with graphite allows for the reduction of the surface resistance and has a key impact on the surface resistance of glass-reinforced composites. The use of carbon fabric as a reinforcement additionally lowers the surface resistance, which is related to its properties. Carbon fiber is a good conductor.

### 3.3. Strength Results

Figure 11, Figure 12, Figure 13 and Figure 14 show the results of flexural strength calculated according to Formula (6) (Figure 12), flexural modulus determined with (7) (Figure 13), and deformation (Figure 14). The stress-strain curves for S1 are presented in Figure 11. The presented results are the arithmetic mean of 5 measurements. With the large discrepancy in the obtained results, the tests were repeated.

The structure S4 (carbon reinforcement) was characterized by the highest flexural strength. It is related to the properties of the reinforcement material. The average flexural strength is 518 MPa, and the standard deviation is 10.8 MPa (approx. 2%), which proves that the fabric is well impregnated. The structure S3 reinforced with a glass mat showed the lowest flexural strength. On the other hand, the highest standard deviation was recorded for the S2 structure, in which hybrid reinforcement was used: glass fabric–glass mat. Diversified properties of the reinforcement disrupted the interaction between the fabric and mat, which resulted in the highest standard deviation of 11%. The structure S2 shows a lower flexural strength by approx. 20% than S1 and at the same time higher by approx. 20% than S3. Therefore, it can be concluded that the value of the flexural strength is the result of the two structures S1 and S3.

S4 structure with carbon fiber reinforcement showed the highest flexural modulus. It is also a result of the properties of the reinforcement material. The structure of S4 shows the smallest standard deviation, approx. 2%. It follows that the designed structure and the developed production technology allowed to obtain a high module. The lowest modulus was shown by composites with glass mat reinforcement—13.2 GPa. The highest standard deviation was for the structure S2—5.5%. It is related to the diverse structure of glass fabric–mat. These materials exhibit different stiffness, which causes laminar skids leading to the failure of the composite.

The structures reinforced with carbon and glass fabrics showed the lowest deformations. For S4, the deformation is 2.03%. The difference between the deformation of the structure S1 and S4 was approx. 25%. It is related to the stiffness, which was confirmed by the results of the flexural modulus. The difference in flexural modules between the structures S1 and S4 was approx. 55%. S4 structure showed the smallest scatter of results. The composite with the S3 structure showed the highest deformation and the highest standard deviation. The structure S2 was characterized by an intermediate strain between S1 and S3, which was confirmed by the flexural strength and modulus. It should be noted that the properties are also affected by the modification of the matrix material with graphite [7].

Based on the properties of the specified flexural strength (Figure 11) and density (Figure 5), the specific flexural strength (Figure 15) was determined.

S4 and S1 structures had the highest specific strengths, which is the result of the previously defined properties. The difference between the flexural strength of the S4 and S1 structures is approx. 15%, while the specific strength is approx. 35%. To verify the quality of the composites with the examined structures, interlayer shear tests were carried out according to the procedure described in Section 2.4. The results are shown in Figure 16.

The highest interlayer shear strengths were demonstrated by composites with S1 and S4 structures of approximately 28 MPa. Composite S2 showed a minimally lower strength, which is undoubtedly related to the manufacturing process. The higher the weight of the fabric, the more difficult it is to saturate it. In the case of S2, shear was observed at the fabric-mat interface, which is related to the difference in the stiffness of the materials. The S3 structures were characterized by the lowest value and crack in compression connected with the interlayer wall according to ISO 14125, insoluble in the double copper wall according to ISO 14130. The structures S1, S2, and S4 showed the failures characteristic of interstitial shear according to ISO 14130: S1 and S4 single shear, an S2 multiple shears. It is also related to the reinforcement layout and the number of layers.

### 3.4. Enclosures Design and Pressure Tests

Based on the conducted research, the physical model of enclosures of the electrical device was designed. For economic reasons, the S1 structure (glass fabric) was adopted. The enclosure wall thickness was 7 mm ± 2 mm. The 500 mm × 500 mm × 300 mm model is shown in Figure 17. The lid was made of steel S335 (Metalex sp. z o.o., Aleksandrów Łódzki, Poland) with a thickness of 10 mm. To ensure an appropriate connection between the lid and the enclosures, a pressure frame made of a hot-rolled angle of 50 mm × 50 mm × 5 mm by S235JR steel was used (Metalex sp. z o.o., Aleksandrów Łódzki, Poland). The whole was screwed with 20 M10 class 10.9 screws, (Elgo sp. z o.o., sp.j., Mazańowice, Poland).

The enclosure was made by hand-up lamination with a vacuum (Figure 18a). In addition, a plate was used to facilitate the formation of the bottom. Before verification tests, the enclosure was conditioned at a temperature of 20 °C for 336 h (2 weeks). Sikaflex–221 (Sika AG, Baar, Swiss) was used to seal the structure (Figure 18b) and bolted joint (Figure 18c). The view of the finished enclosures for research is presented in Figure 18c.

The test stand (Figure 19) consisted of a water tank, an IBO PR-AUTO pump with a capacity of 174 l min, maximum pressure of 60 bar, equipped with a manometer, filtering system, and supply hoses (IBO, Ferrara, Italy).

The pressure tests carried out showed that the proposed enclosure S1 structure and wall thickness of 7 mm withstands a pressure of 8 bar. During the tests, a minimal deformation of the composite was observed in the central area of the walls, but there were no cracks, leaks, or even dew. The tests showed that the weakest point of the enclosures is the connection of the composite with the metal lid, where the leaks occurred. The internal pressure of 8 bar gives a load of 196.13 kN on the lid, which causes deformation and leakage in the flange area during tests.

## 4. Conclusions

Based on the conducted research, the following was concluded:The S4 structure is characterized by the highest tested thermal properties, surface resistive, flexural strength, and modulus. It is related to the 800 g/m^2^ quadriaxial reinforcement material used.Interlayer shear strength of S4 structure is 19 MPa lower than that of composite S1. This is due to the impregnation of the fabric. Glass fabrics of 550 g/m^2^ are much easier to impregnate than 800 g/m^2^, using the technology of hand-up lamination with a vacuum. The structure S1 also showed a higher strain than S4, which is a better solution assuming the target application.The lowest values of the glass composites (structures S1, S2, S3), flexural strength, and modulus are characteristic of the composite with the S3 structure. Composite S2 shows properties that average the structures S1 and S3. The structure S3 was characterized by the lowest value of inter-layer shear. It is related to the layout, type, and several number layers of reinforcement.Composites with S1, S2, and S3 structures are characterized by lower values of the thermal conductivity coefficient and higher surface resistivity than S4. It is related to the properties of glass, which plays an important role here. Graphite introduced into the vinyl ester resin significantly increases the specific heat and thermal conductivity and lowers the surface resistivity.The developed procedure for determining the specific heat based on the rule of mixtures allows approximating the value of specific heat by approx. 10% for composites reinforced with fabrics. For composites reinforced with fabric and mat or only mat, the approximate value is approx. 5%.The designed and made an explosion-proof enclosure for electrical devices made of S1 composite with a wall thickness of 7 mm, withstands an internal pressure of 8 bar. The composite showed no damage. Leaks having a significant impact on pressure resistance were observed at the connection enclosure–lid. This requires the design of a different, more airtight joint sealing system.

## Figures and Tables

**Figure 1 materials-15-05190-f001:**
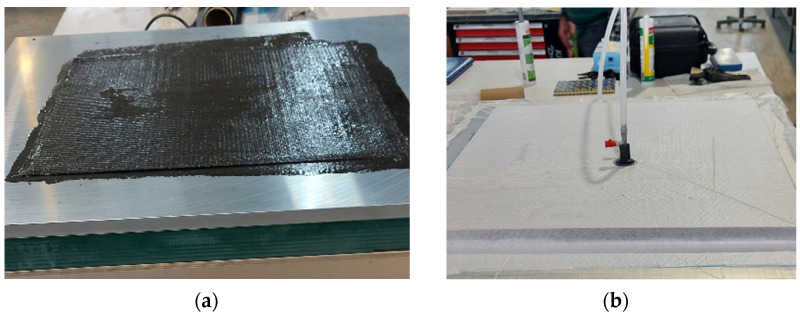
Composite lamination (**a**) with vacuum support (**b**).

**Figure 2 materials-15-05190-f002:**
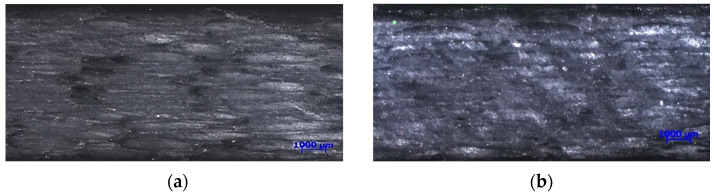
Microscopic images of S1 (**a**), S2 (**b**), S3 (**c**) and S4 (**d**) structures.

**Figure 3 materials-15-05190-f003:**
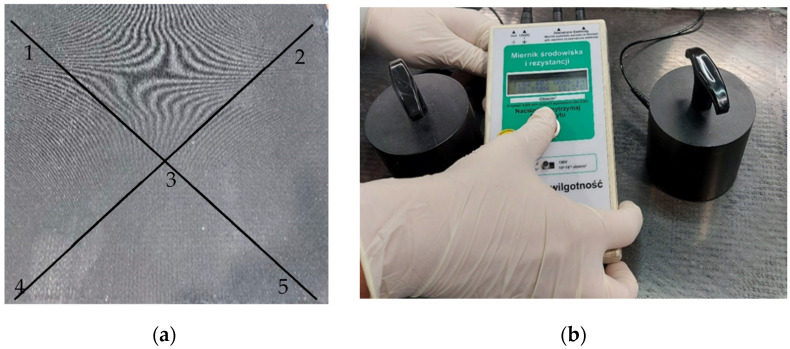
View of the samples with measurement points (**a**). Device for measuring surface resistivity (**b**).

**Figure 4 materials-15-05190-f004:**
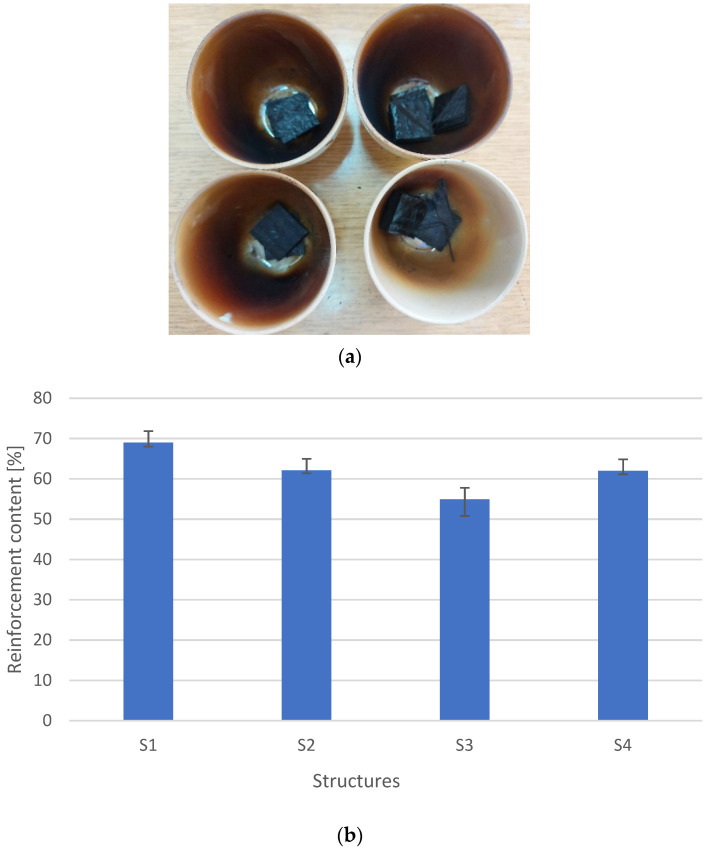
Reinforcement content (**a**) view after pyrolysis, (**b**) graph on the dependence of different structures.

**Figure 5 materials-15-05190-f005:**
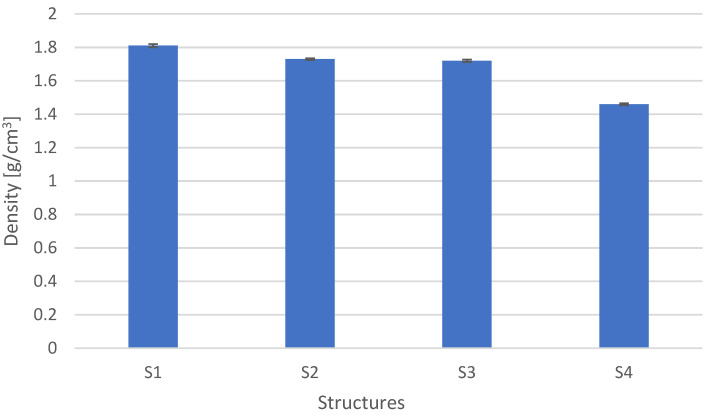
Density for different composite structures.

**Figure 6 materials-15-05190-f006:**
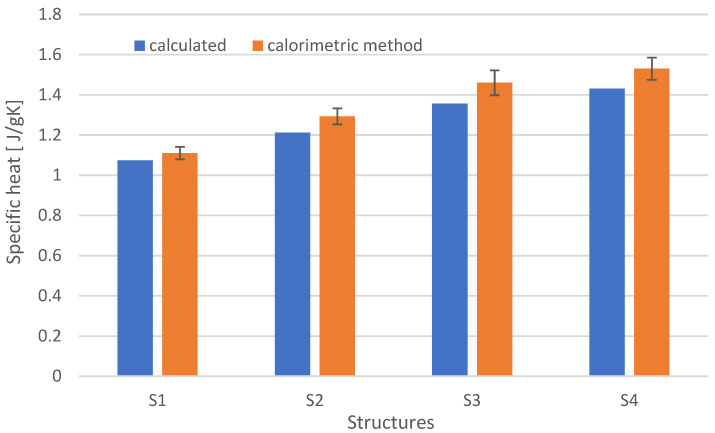
Specific heat for different composite structures.

**Figure 7 materials-15-05190-f007:**
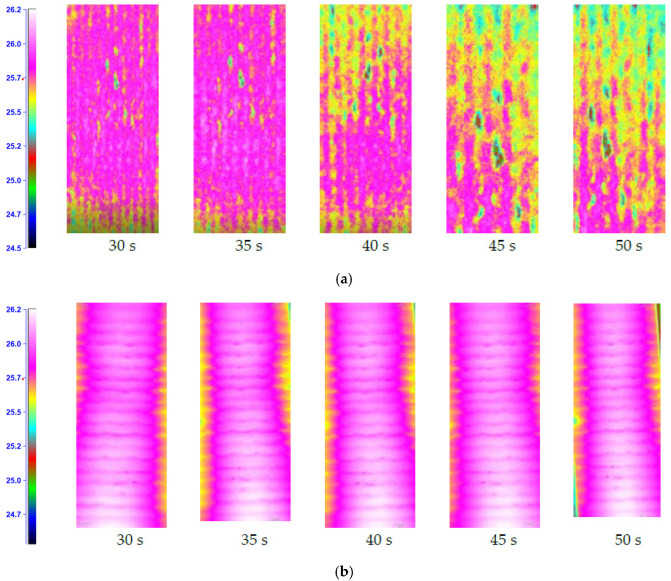
Temperature distribution maps for S4 (**a**) and S1 (**b**) structure composite.

**Figure 8 materials-15-05190-f008:**
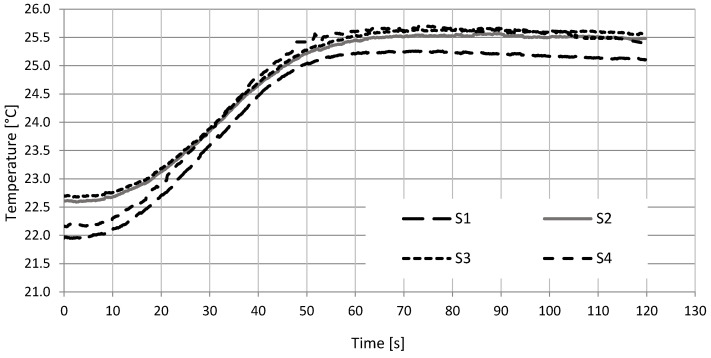
Temperature change during heating.

**Figure 9 materials-15-05190-f009:**
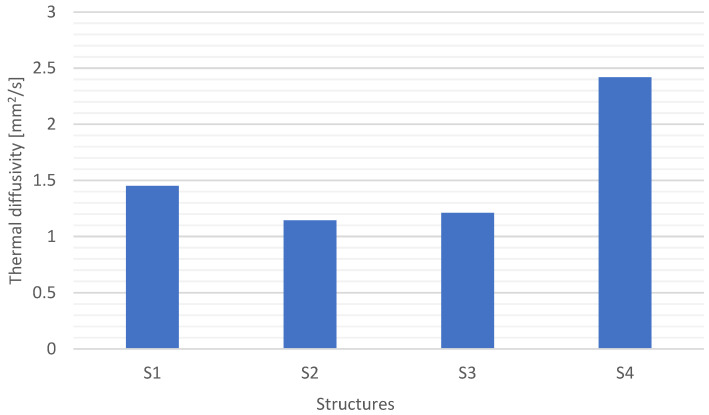
Thermal diffusivity for different composite structures.

**Figure 10 materials-15-05190-f010:**
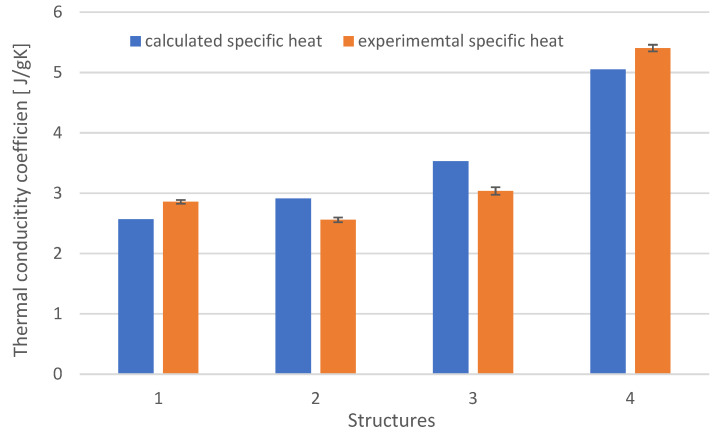
Thermal conductivity coefficient for different composite structures.

**Figure 11 materials-15-05190-f011:**
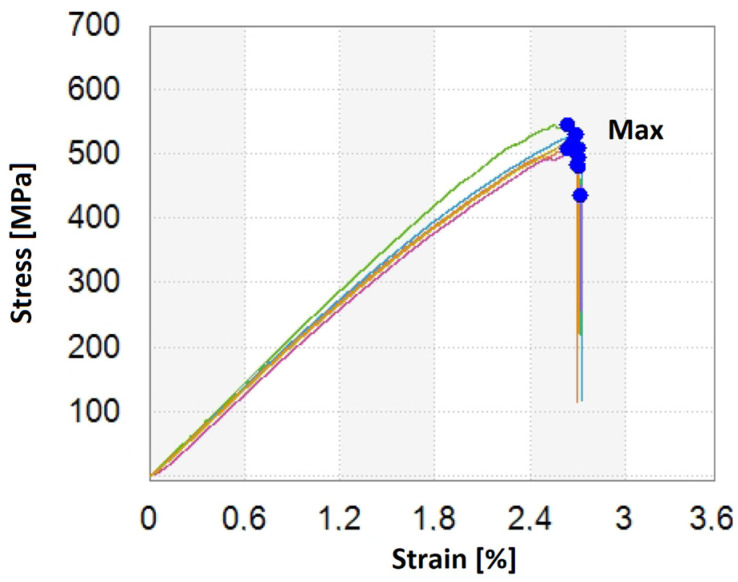
Stress-strain curves for S1 structure.

**Figure 12 materials-15-05190-f012:**
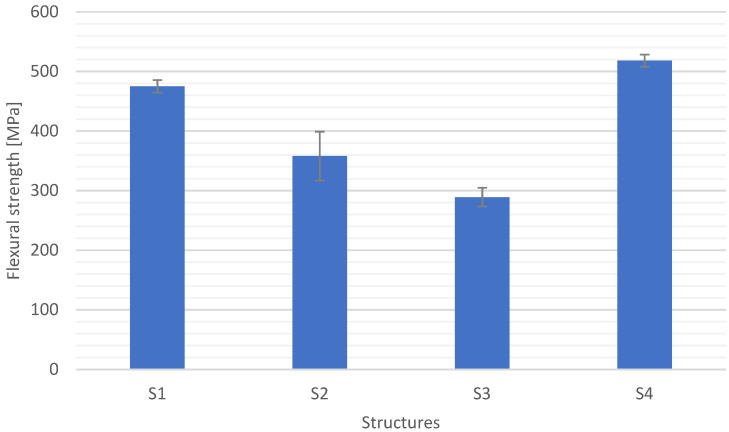
Flexural strength for different composite structures.

**Figure 13 materials-15-05190-f013:**
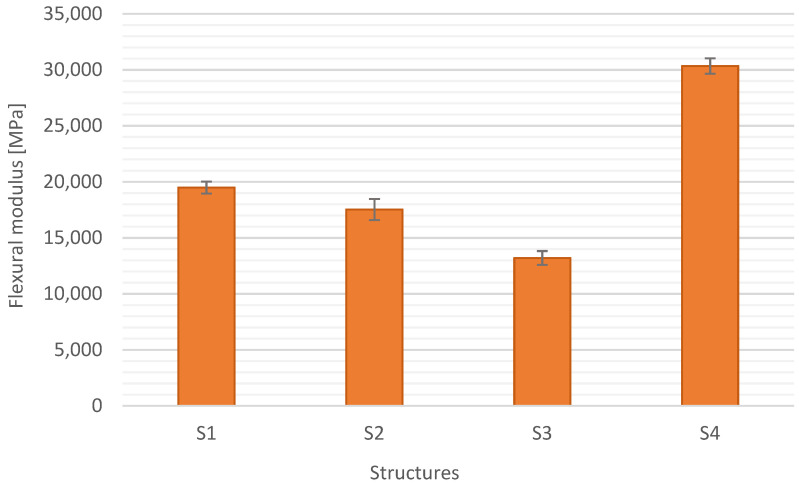
Flexural modulus different composite structures.

**Figure 14 materials-15-05190-f014:**
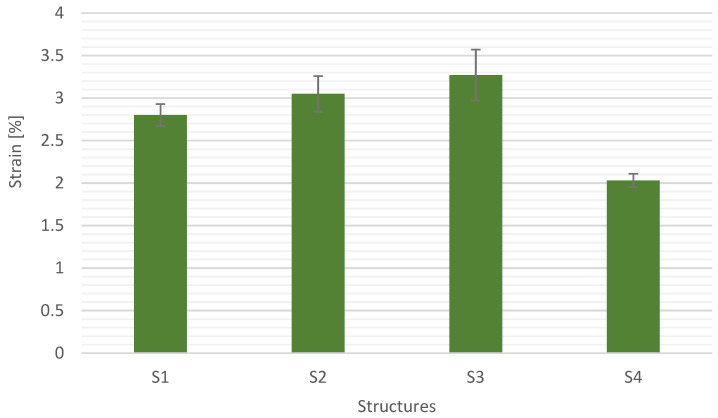
Strain for different composite structures.

**Figure 15 materials-15-05190-f015:**
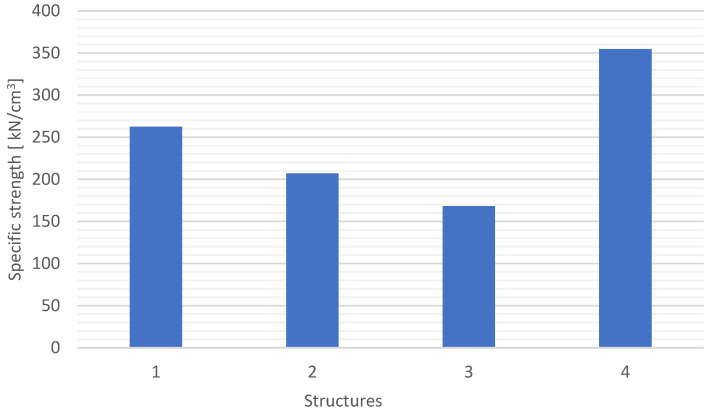
Specific flexural strength for different composite structures.

**Figure 16 materials-15-05190-f016:**
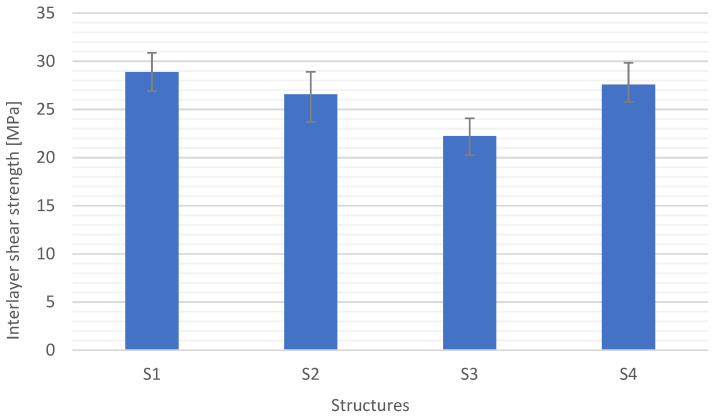
Interlayer shear strength for different composite structures.

**Figure 17 materials-15-05190-f017:**
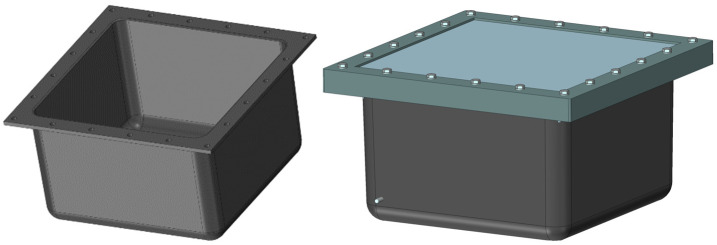
Enclosure model.

**Figure 18 materials-15-05190-f018:**
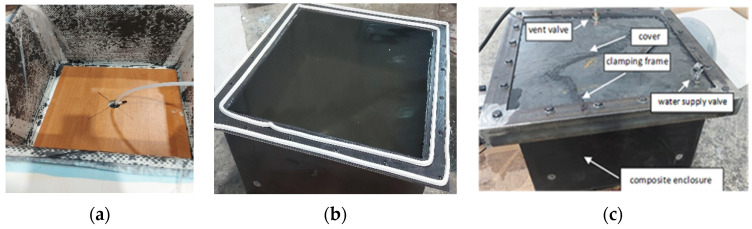
Enclosure physical model. Technology (**a**), seal (**b**) prepared enclosure for testing (**c**).

**Figure 19 materials-15-05190-f019:**
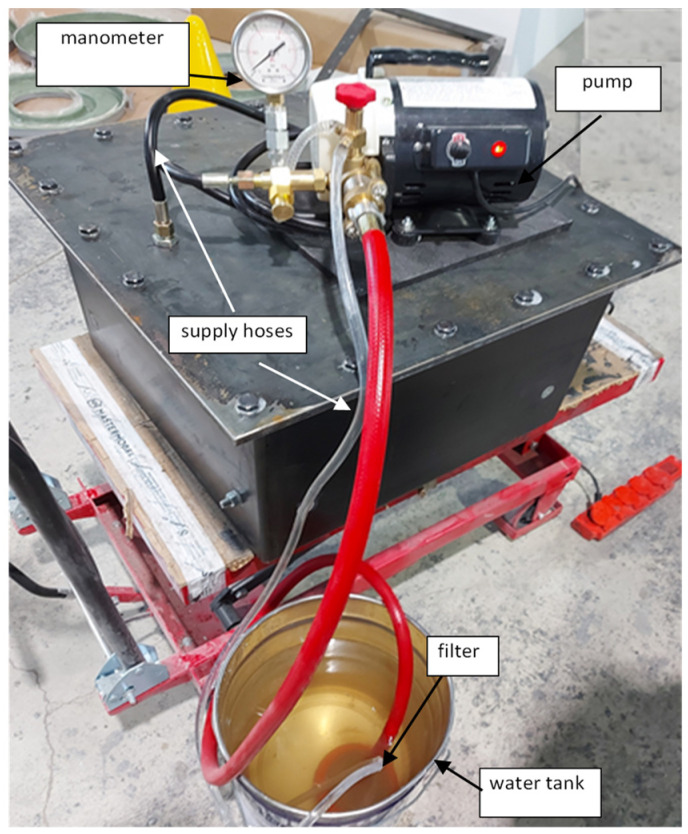
Pressure test stand.

**Table 1 materials-15-05190-t001:** Composite structures.

S1		S2		S3		S4	
Material	Number of Layers	Material	Number of Layers	Material	Number of Layers	Material	Number of Layers
Glass triaxal 550 g/m^2^ (0/45/90)	13	Glass triaxal 550 g/m^2^, (0/45/90)	3	Glass mat 450 g/m^2^	16	Carbon quadrax 800 g/m^2^ (0/−45/90/45)	9
		Glass mat 450 g/m^2^	7				
		Glass triaxial 550 g/m^2^, (0/45/90)	3				
	Σ13 ~7 mm		Σ13 ~7 mm		Σ16 ~7 mm		Σ9 ~7 mm

**Table 2 materials-15-05190-t002:** Properties of matrix [7].

Property	Standard	Unit	Modified POLIMAL-VE-11 MAT
Density	ISO 1183-1	g/cm^3^	1.55
Gel time at 25 °C	ISO 2535	min	25
Flammability class	EN 60695-11-10:2014-02	-	V0
Flexural strength	ISO 178	MPa	50.61
Flexural modulus	ISO 527	MPa	5271.6
Strain at break	ISO 527	%	1.34
Specific heat	ISO 11357-4:2013	J/gK	2.46
Thermal conductivity coefficient	-	W/mK	5.64
Surface resistivity	IEC 61340-2-3	Ω	5.19 × 10^3^

**Table 3 materials-15-05190-t003:** Surface resistivity for different composite structures.

Structures	Surface Resistivity, Ω
S1	3.36 × 10^3^
S2	3.15 × 10^3^
S3	2.85 × 10^3^
S4	<10^3^

## Data Availability

Not applicable.

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
