# Peer review of "Influence of the Reinforcement Structure on the Thermal Conductivity and Surface Resistivity of Vinyl Ester Composites Used on Explosion-Proof Enclosures of Electrical Equipment"

_materials, 2022, doi:10.3390/ma15155190_

Round 1

Reviewer 1 Report

Title: Influence of the reinforcement structure on the thermal conductivity and surface resistivity of vinyl ester composites used on explosion-proof enclosures of electrical equipment

The referee would like to recommend this work to minor revision and to be published after consideration according to the comments below:

1.      The writing of this paper needs to be improved and polished. Some clumsy and neglectful expressions can be found.

2.      Please add more current papers in the literature and improve introduction section. Some interesting papers related to the topic of this manuscript could be:

·           Size-dependent nonlinear instability of shear deformable cylindrical Nano panels subjected to axial compression in thermal environments, (2017), Microsystem Technologies  23(10), 4717-4731.

3.      Rule of mixture is for composites with long and homogeneous reinforcements. Please explain how you can use this rule (Eq. 4).

4.      Please explain why did you consider the shear stress correction factor to be 4/3 (Eq. 7)?

5.      For good results, any test should be performed at least three times, which is not the case in this study. Please explain.

Author Response

Dear Reviewer,

Thank you for the work you put into the review, comments, and suggestions that improved the clarity of the work. Please find attached detailed answers.

Reviewer 2 Report

The author in the manuscript has evaluated the influence of structure on thermal properties and surface resistance of composites. The manuscript is well written. I recommend the publication of the manuscript. Few things which should be taken care of are as below:

1) Was any SEM image recorded to look into the uniform distribution of the carbon ?

2) Comparison of temperature map would be ideal, such as S3 vs S4 instead of just S4.

Author Response

(The authors gave the same response as above.)

Reviewer 3 Report

The manuscript entitled “Influence of the reinforcement structure on the thermal conductivity and surface resistivity of vinyl ester composites used on explosion-proof enclosures of electrical equipment” has been reviewed. The results are helpful. However, the manuscript needs to be well revised before acceptance. Detailed comments are as follows:

1.      The manuscript was not well prepared. The quality of English and writing style should be better.

2.      The full names of some abbreviations, such as RTT and RTG should be given where the abbreviations first appeared.

3.      There are too many typo errors in the manuscript. Units should be separated from the numerical value by a space. No space should be put after /. o C should be oC. pay attention the subscript form of some numbers, such as g/cm3. Please recheck your manuscript.

4.      Pay attention to the cited authors. Du should be Du and Fang. Jaswal should be Jaswal and Gaur. El-Hage should be El-Hage et al. Xiao-Yi Zhou et all [6] should be Zhou et al. [9].

5.      2. Materials for Research and Methodology should be just before 2.1. Materials for Research since the paragraph “The work aimed at assess the impact……” belongs to Introduction.

6.      The number of duplicates of each sample for the figures with deviation bars should be given in the tests of Methodology section.

7.      Please unify h and hours.

8.      UL 94 V0 should be V-0.

9.      The values of surface resistivity should be expressed in the form of Table 2.

10.  References should be revised as per guide for authors of Materials. Pay attention to unifying the capitalization of first letters of paper titles. Some references, such as [1] is not complete.

Author Response

(The authors gave the same response as above.)

Reviewer 4 Report

The influence of structure (type and material) on thermal properties (thermal conductivity, diffusivity) and surface resistance of composites used for explosion-proof enclosures of electrical devices were evaluated in the manuscript named Influence of the reinforcement structure on the thermal conductivity and surface resistivity of vinyl ester composites used on explosion-proof enclosures of electrical equipment (Manuscript ID: materials-1790408). Which aims to design and manufacture explosion-proof enclosures for electrical devices made of polymer composites. While, some questions should be well illuminated before the manuscript being further processed.

1.        The results and the corresponding conclusions should be displayed in the Abstract section to offer readers an overall look of the research work.

2.        It was stated in the title that “the thermal conductivity and surface resistivity of vinyl ester composites” were studied, while in the introduction section, the thermal conductivity of composites was not comprehensively reviewed, and how the thermal conductive fillers in this work was selected is not well deduced.

3.        Following the last question, the thermal conductive fillers were mentioned in the fourth paragraph in the introduction, thermal conductive fillers such as graphite (Progress in Polymer Science 61 (2016) 1-28; International Journal of Heat and Mass Transfer 155 (2020) 119853), carbon materials, carbon nanotube, and graphene have been widely utilized to enhance the thermal conductivity. There should be a comprehensive review on these fillers to offer a reasonable selection criterion, because the graphite was mentioned in the introduction section while the carbon quadriaxial fabric, glass based fillers were selected, which is strange and unreasonable.

4.        The thermal conductivity was obtained using equation 1, while it was weird that the corresponding thermal diffusivity and specific heat were not directly measured experimentally, which makes the data of thermal conductivity unpersuasive. Since the thermal diffusivity and specific heat are highly dependent on the structure of the composite materials, which is not accurate to theoretically calculate. To obtain accurate data of the thermal conductivity, the thermal diffusivity and specific heat could be obtained using flash test test and DSC, the specific method is recommended in the reference (Membranes 2022, 12, 222).

5.        The microstructure and morphology of the fabricated composite materials should be present in the paper, since the topic this work is focused on the effect of filler structure on the thermal conductivity, but no evidence could support that composite with different filler structures were obtained in this work.

6.        The stress-strain curves of these composite materials and the corresponding fracture surface of the mechanical test sample should be present to support the conclusion of the mechanical test.

7.         It is difficult to understand the layer described in table 1, the suggestion is to offer a schematic illustration on the fabrication of the composite materials.

8.        It should be well explained why different filler content was added in these four composite materials in Fig.3?

9.        The conclusion should be clear and concise, in which the key point and novelty should be summarized rather than wordy description.

Author Response

(The authors gave the same response as above.)

Round 2

Reviewer 1 Report

Accepted

Author Response

Dear Reviewer, Thank you for all the remarks and comments that have influenced the quality of the article.   Best regards

Reviewer 3 Report

Most comments have been revised. The manuscript can be accepted if the following comments are considered:

1.      There are still some typo errors in the manuscript. Units should be separated from the numerical value by a space, especially in Table 1. Please double-check your manuscript.

2.      UL 94 V0 should be V-0 in Table 2.

Author Response

(The authors gave the same response as above.)

Reviewer 4 Report

Some improvememts were displayed in the updated version. While, the key parameter, thermal conductivity, especially the specific heat and thermal diffusivity obtained by the rule of mixture in this work is not convincing. And the conclusion in this work is unable to offer a useful reference for other researcher. 

Author Response

Dear Reviewer,

Thank you for your comments. Sorry, the significant improvements made did not affect the quality of the article in your opinion. Referring to the current comments, we would like to inform you:

  • The thermal diffusivity was determined not by the mixtures rule, but by the Parker method with the use of thermovision. The measuring principle is described in chapter 2.2. A detailed description of the Parker thermal diffusivity determination is found in the cited paper [18] Parker, W.J.; Jenkins, R.J .; Butler, C.P .; Abbott, G.L. Flash Method of Determining Thermal Diffusivity, Heat Capacity, and Thermal Conductivity. Journal of Applied Physics 1961, 32, 1679-1684, doi: 10.1063 / 1.1728417.
  • In order to check the described method, the specific heat was additionally determined using the calorimetric method. It is related to the geometry and structure of the sample. Samples of various thicknesses can be examined in the calorimeter. The results are summarized in Fig. 6. The differences between the determined heat of the structures S1 and S4 amount to approx. 10%, in the case of structures S2 and S3 approx. 5%.
  • Conclusions were corrected

Please find attached the completed and corrected work. The last corrections have been marked in blue.

We hope that the introduced changes will meet your expectations. 

Best regards
